# “Back Health 24/7/365”—A Novel, Comprehensive “One Size Fits All” Workplace Health Promotion Intervention for Occupational Back Health among Hospital Employees

**DOI:** 10.3390/ijerph21060772

**Published:** 2024-06-14

**Authors:** Timothy Hasenoehrl, Margarete Steiner, Felix Ebenberger, Philipp Kull, Julia Sternik, Lukas Reissig, Galateja Jordakieva, Richard Crevenna

**Affiliations:** 1Department of Physical Medicine, Rehabilitation and Occupational Medicine, Medical University of Vienna, Waehringer Guertel 18-20, 1090 Vienna, Austria; 2Division of Anatomy, Center for Anatomy and Cell Biology, Medical University of Vienna, Waehringer Straße 13, 1090 Vienna, Austria

**Keywords:** back pain, manual material handling, office work, weight lifting, core stability, education, compensatory exercises

## Abstract

Background: Projects for workplace health promotion (WHP) for back pain traditionally focus exclusively on work-related but not on leisure-time stress on the spine. We developed a comprehensive WHP project on the back health of hospital workers regardless of the physical characteristics of their work and compared its effects on sedentary and physically active hospital workers. Methods: Study assessments were carried out before and six months after participation in the WHP intervention. The primary outcome parameter was back pain (Oswestry Disability Index, ODI). Anxiety (Generalized Anxiety Disorder-7), work ability (Work Ability Index), depression (Patient Health Questionnaire-9), stress (Perceived Stress Scale-10), and quality of life (Short Form-36) were assessed via questionnaires as secondary outcome parameters. Physical performance was measured via the 30 seconds Sit-to-Stand test (30secSTS). Results: Sixty-eight healthcare workers with non-specific back pain were included in the evaluation study of the WHP project “Back Health 24/7/365”. After six months, back pain, physical performance, and self-perceived physical functioning (SF-36 Physical Functioning subscale) improved significantly in both groups. Not a single parameter showed an interaction effect with the group allocation. Conclusions: A comprehensive WHP-intervention showed significant positive effects on hospital workers regardless of the physical characteristics of their work.

## 1. Introduction

Nonspecific low back pain (NSLBP) is a common health problem worldwide. It has been known already for a long time that healthcare workers—in particular nurses—are at an increased risk of developing work-related back pain [1]. Depending on the occupational group, their lifetime prevalence rate of problems in the lower back area is up to 85% [2]. This not only negatively impacts healthcare systems from the economic perspective [3,4] but also negatively impacts the healthcare system itself [5].

For this reason, workplace health promotion (WHP) projects have been launched around the world, many of them with the aim of counteracting the development of NSLBP or treating existing NSLBP [6].

WHP projects have proven to be effective against work-related NSLBP, particularly among healthcare workers [7,8].

However, traditionally, in WHP projects, office workers are taught to be more physically active and counteract the musculoskeletal consequences of sedentary behavior [9], while heavy workers receive manual material handling training [10,11].

However, once a WHP project is designed—particularly in healthcare—we believe that specific distinctions need to be taken into account:

First, NSLBP develops from the two extremes of the physical activity continuum: NSLBP can be caused either by excessive physical stress on the spine [12] or by physical inactivity and sedentary behavior [13]. This means that very physically active workers, e.g., nurses, who carry patients and thereby put unphysiological strain or overload on their spine, can be affected by NSLBP [14], as can administrative employees who carry out sedentary office work [9].

Second, when designing and planning a WHP project, at the individual level, physical activity behavior at work does not necessarily have to match physical activity behavior during leisure time [15]. It may be that someone who has a sedentary office job has a very physically demanding private life, and someone who has a physically demanding job can lead a sedentary private life. Therefore, in our opinion, it is not possible to ignore the physical leisure activity of the participants in the design of a health promotion project at work.

In our opinion, if an occupational health project is designed only for a specific workplace—be it a sedentary or heavy lifting job—it may not reflect the complex needs at an individual level. For this reason, we have designed a comprehensive project to promote spinal health in the workplace: “Back Health 24/7/365”. This WPH concept includes both a theoretical and a practical educational measure and takes into account both the dimension of over- and underloading of the spine. The aim of this WPH project was to teach hospital employees both theoretically and practically, how to keep their spine healthy, not only particularly at work but generally in all (in-) activities of daily living. 

In the current manuscript, we present the evaluation of this WPH project. 

We hypothesized that this novel comprehensive “one-size-fits-all” approach will improve the back health of all hospital employees equally, regardless of the physical demands of their job.

## 2. Materials and Methods

### 2.1. Participants

The WPH promotion project “Back Health 24/7/365” took place at the Vienna General Hospital, Austria, one of the largest university hospitals in Europe with over 1700 beds. Together with the employees of the Medical University of Vienna and the operating company, around 15,000 people in countless different jobs are employed at the location and were invited to participate in this WPH project and the associated evaluation study.

All employees of the General Hospital of Vienna, Austria, the Medical University of Vienna, Austria, and the operating company, VAMED, were invited via online newsletter and the works council of their respective employer. Employees could participate in the WPH project without participating in the evaluation study. Inclusion criteria were existing NSLBP and medical suitability for participation in a physically active training course. Exclusion criteria were specific back pain, musculoskeletal surgery (planned during the course of the study), and insufficient language skills.

The study protocol was approved by the ethics committee of the Medical University of Vienna, Austria, (EK1074/2023) and was performed in accordance with the ethical standards laid down in the 1964 Declaration of Helsinki. All participants gave their informed consent prior to their inclusion in the study. 

### 2.2. Intervention

The WPH project consisted of four parts, one theoretical video training and three practical workshops. The video training consisted of four short videos in which experts presented their knowledge of NSLBP in their specific field. The anatomy of the spine was presented by a specialist in anatomy, how mental health can affect NSLBP was presented by a clinical psychologist, the effects of nutrition and obesity on spinal health were presented by a nutritionist, and the medical treatment guidelines for NSLBP were presented by a physiatrist who is also a specialist in pain medicine. 

The three practical workshops were led by an expert in sports and applied medical sciences. In the first workshop, participants received manual material handling training with particular emphasis on basic weightlifting techniques such as spinal bracing and deadlifts. In the second workshop, participants were taught specific compensatory exercises for prolonged sitting, e.g., strengthening the back extensors or how to stretch tight muscles, and in the third workshop, the participants received general training in core stability and mobility exercises (Appendix A).

The respective training courses were offered to the participants in small groups of a maximum of 10 participants over 10 weeks between April and June 2023 at their own discretion. After completing all of the training, all participants received a training booklet with the most important information and all the exercises taught.

### 2.3. Assessments and Outcome Parameters

Participants in the evaluation study were assessed at baseline (BL) and 6 month (FU) after completion of their training. The primary outcome parameter was the Oswestry Disability Index (ODI), a questionnaire which is used to assess functional disorders caused by back pain [16]. Sample size for the interaction effect with an alpha-error probability of 0.05, a beta-error probability of 0.8 and an estimated effect size of 0.25 resulted in a sample size of 34 participants [17]. 

Several validated questionnaires were used as secondary outcome parameters: The Work Ability Index (WAI), which is a measuring instrument for recording the work ability of employed people [18], the Generalized Anxiety Disorder-7 (GAD-7), which was designed to identify patients with generalized anxiety disorder and to record the symptom severity of general anxiety [19], the Patient Health Questionnaire (PHQ-9), a screening instrument for diagnosing of depression [20], the Perceived Stress Scale (PSS-10), which measures the subjective experience of stress [21], the Fear Avoidance Beliefs Questionnaire (FABQ-D), which records patients’ attitudes regarding the influence of physical activity and work on back pain [22], and the Short Form-36 (SF-36), a generic measurement instrument for recording the health-related quality of life of patients [23]. Additionally, physical performance was measured via the 30 Seconds Sit-to-Stand Test (30secSTS) which is used to estimate the strength endurance of the lower extremities and describes how often the test subject can get up from a chair and sit down again in a period of 30 s [24]. 

### 2.4. Statistical Analyses

Based on the information obtained during the baseline assessment, the sample was divided into two groups: those employees with a physically inactive/sedentary workplace (INACT) and the others with a physically active workplace (ACT). A mixed models analysis of variance (ANOVA) was conducted for the effect of time and the interaction effect of time over group to discover potential differences between the groups. Homogeneity of variances was tested via the Levene test. Baseline characteristics were compared utilizing independent t-tests to reflect potential differences between the groups at baseline.

## 3. Results

One-hundred employees participated in the WHP intervention “Back Health 24/7/365”. Thirty-two only wanted to participate in the WHP promotion project but not in the evaluation study. Therefore, 68 hospital workers were included in the evaluation study and participated in all training courses. Two participants dropped out of the study for medical reasons unrelated to the WHP intervention, and one participant decided to drop out of the study because she did not want to answer all of the questionnaires. Twenty-one participants did not complete FU measurements at 6 months, leaving forty-four employees participating in all interventions and study assessments. Study drop-outs were not linked to job profile and were completely random. 

BL assessments included several questions designed to determine participants’ work characteristics related to physical work. Based on this information, 28 of the participants were assigned to the INACT group and 16 to the ACT group. Comparison of baseline characteristics between ACT and INACT did not show any significant group differences at BL (Table 1). 

In the primary outcome parameter, the ODI, mixed-models ANOVA showed a significant effect of the factor time (F(1, 42) = 7.042, *p* = 0.011) but no interaction effect (F(1, 42) = 0.004, *p* = 0.951) (Figure 1). Similarly, the 30secSTS also increased significantly over time (F(1, 42) = 12.409, *p* = 0.001) but independently from the group allocation (F(1, 42) = 0.430, *p* = 0.516) (Figure 2). Also, anxiety, measured via GAD-7, decreased significantly over time (F(1, 42) = 4.656, *p* = 0.037) but did not show an interaction effect between time and group (F(1, 42) = 0.668, *p* = 0.418) (Figure 3). In the SF-36, the subscale Physical Functioning showed significant increases over time (F(1, 41) = 5.936, *p* = 0.019) which was independent from the group allocation (F(1, 41) = 0.001, *p* = 0.977) (Figure 4). All other parameters showed no significant effects, neither for time nor for an interaction effect between time and group (Table 2 and Appendix A).

## 4. Discussion

The results of the evaluation study of our WHP project “Back Health 24/7/365” showed that, regardless of the physical activity characteristics of their workplace, participants showed significant improvement in their lower body back pain, strength endurance, anxiety and self-perceived physical performance 6 months after participating in the WHP training. Since the aim of the WHP Back Health 24/7/365 project was to demonstrate that both physically working and sedentary hospital workers would equally benefit from a comprehensive educational approach to back health, it can be concluded that our primary hypothesis can be supported. 

This is particularly relevant because the WHP project “Back Health 24/7/365” is new from a scientific perspective. To our knowledge, there have been no WHP intervention projects for back pain that have included both sedentary or “office workers” and physically active workers and provided them with comprehensive training for all activities of daily living, not just for their specific workplace [25]. 

Interestingly, the literature about the efficacy of different WHP interventions has been inconsistent. While Verbeek et al. (2012) [26] concluded that manual material handling trainings were inefficient in preventing low back pain in physical workers, WHP interventions comprising exercise training consistently showed positive outcomes [27,28,29]. Considering that professional weightlifters do not have a particularly high rate of spinal disorders compared to other professional athletes [29], and that weightlifting per se has a low incidence of back injuries compared to other sports [30], it would seem that there is a discrepancy between spinal load management in weightlifting and what is taught in manual material handling training courses. This would explain why our comprehensive intervention in contrast to Verbeek et al. (2012) [26], who systematically reviewed studies that utilized manual material handling interventions only, showed significant improvements. Studies included in this systematic review were all purely manual material handling interventions for physical workers and, unlike our project, back health outside the workplace was not taken into account [26]. This is a massive blind spot in this field of WHP research and it shows that our comprehensive approach which was based to a large extent on sport scientific knowledge is not only novel but highly relevant for workplace back health. 

A considerable extent of WHP projects for sedentary workplaces primarily focusses on reducing sitting time and promoting physical activity to increase health and work ability [30,31,32,33]. To our knowledge, WHP intervention trials that provided exercises that were particularly designed to counteract a sedentary workplace are scarce [34,35,36]. A sedentary workplace stresses the spine differently than a blue collar workplace [37] and typically results in a number of specific changes to the musculoskeletal system [38,39,40]. These unintentional adaptations should be addressed specifically and not just by increasing physical activity behavior in general—at least if improving back health is the main goal. This is why we included specifically counteracting exercises to our WHP intervention. 

Apart from the health perspective, the economic perspective of WHP projects is less clear. From an economic perspective, manual material handling training has demonstrated high cost-effectiveness [41], while the other literature shows that WHP for back pain has not had proven cost-effectiveness [42]. As mentioned earlier, this discrepancy might be to a substantial extent based on the fact that up to date it has not been common to consider the leisure-time physical activity behavior into the design of workplace health projects. This is a specific strength of this project as the WHP “Back Health 24/7/365” intervention was never focused solely on the workplace but was always aimed at supporting hospital staff, regardless of the demands of their workplace, to improve their general understanding of spinal health. The results of our study support the design of our WHP intervention. Additional strengths of this study are its comprehensive and novel WHP intervention and the fact that this project was undertaken in one of the largest university hospitals worldwide.

However, this study also has some limitations. First, the number of employees participating in this study is rather small given the number of eligible employees on site. Furthermore, our sample certainly did not include all the professional groups that could be found in a huge university hospital. The first point is correct from a WHP perspective. It is always desirable to reach larger parts of the eligible population. However, the recruitment process revealed systematic limitations in a hospital setting. In particular, we received feedback from the surgical departments that it was simply not possible to take so much time off during working hours in their strict surgical schedule, while in our own department, the WHP offer was limited to consultation hours for employment law and patient insurance reasons. In addition, departments with staff shortages could not send their staff because they could not afford to take even more staff off. From this feedback, we learned that in the future, we will need to find a solution to implement this training differently, e.g., directly in the departments. However, from a scientific perspective, none of these reasons affected the results of our evaluation study and our sample met the previously calculated sample size criteria. And for the second point, we collected information about each participant’s job characteristics at baseline and then carefully divided the sample into the INACT and ACT groups. If there was any uncertainty about the workplace characteristics, the respective participant was contacted personally, and the physical workplace requirements were discussed in detail.

## 5. Conclusions

The evaluation of the comprehensive and novel WHP intervention “Back Health 24/7/365” showed that a comprehensive, low-effort “one size fits all” workplace intervention can significantly impact hospital workers’ back health and physical performance capacity regardless of their physical work characteristics. 

## Figures and Tables

**Figure 1 ijerph-21-00772-f001:**
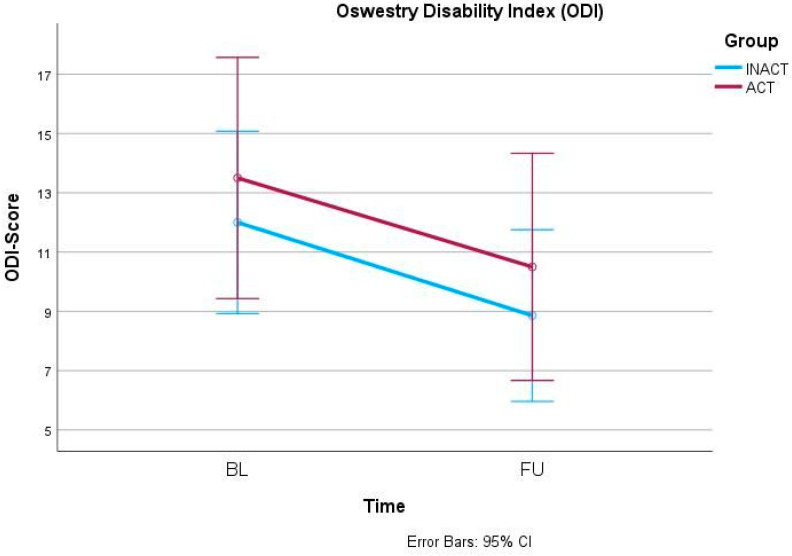
Oswestry Disability Index (ODI) changes from BL to FU.

**Figure 2 ijerph-21-00772-f002:**
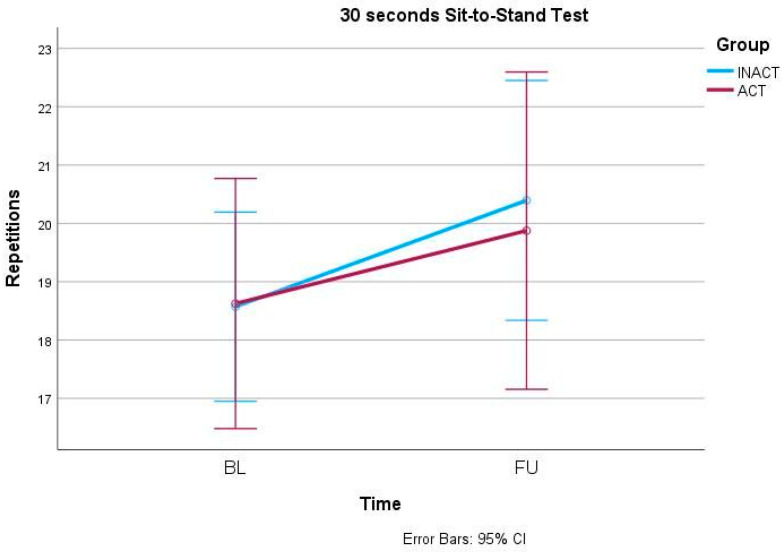
30 seconds Sit-to-Stand Test (30secSTS) changes from BL to FU.

**Figure 3 ijerph-21-00772-f003:**
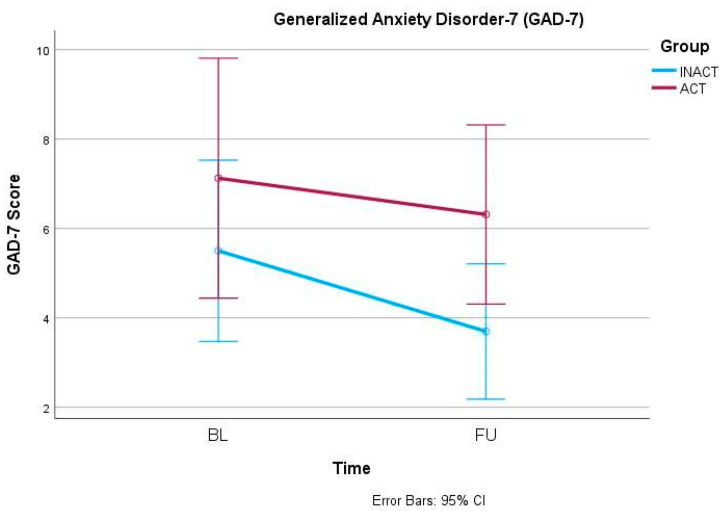
Generalized Anxiety Disorder-7 (GAD-7) changes from BL to FU.

**Figure 4 ijerph-21-00772-f004:**
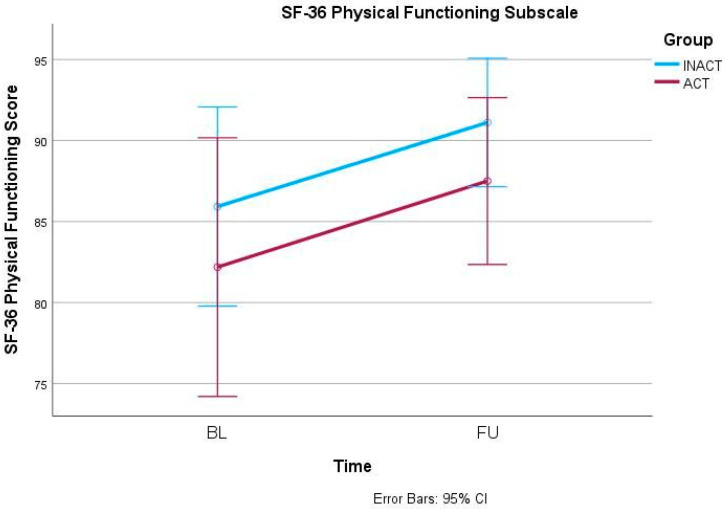
Short Form-36 (SF-36) Physical Functioning Subscale changes from BL to FU.

**Table 1 ijerph-21-00772-t001:** Comparison of baseline characteristics between groups.

Outcome Parameter	Group	N	Mean ± SD	Sig.	Mean Difference	95%CI Lower Limit	95%CI Upper Limit
Age	INACT	28	47.0 ± 8.8	0.626	−1.437	−7.35	4.48
	ACT	16	48.4 ± 10.3				
Height	INACT	28	164.4 ± 30.1	0.573	−4.357	−19.83	11.12
	ACT	16	168.8 ± 6.6				
Weight	INACT	28	70.0 ± 15.3	0.312	−5.277	−15.68	5.13
	ACT	16	75.3 ± 18.4				
Waist-to-Hip Ratio	INACT	27	0.86 ± 0.1	0.231	−0.04	−0.11	0.03
	ACT	13	0.91 ± 0.1				
ODI	INACT	28	12.2 ± 8.3	0.556	−1.500	−6.60	3.60
	ACT	16	13.5 ± 7.5				
30secSTS	INACT	28	18.6 ± 4.1	0.968	−0.054	−2.74	2.63
	ACT	16	18.6 ± 4.4				
WAI	INACT	28	39.6 ± 7.6	0.054	4.357	−0.07	8.79
	ACT	16	35.3 ± 5.8				
GAD-7	INACT	28	5.5 ± 5.9	0.335	−1.625	−4.99	1.74
	ACT	16	7.1 ± 4.2				
PHQ-9	INACT	28	5.0 ± 5.0	0.188	−2.062	−5.17	1.05
	ACT	16	7.1 ± 4.9				
PSS-10	INACT	28	12.3 ± 7.0	0.322	−2.179	−6.57	2.21
	ACT	16	14.5 ± 6.9				
SF-36 Physical Sumscale	INACT	26	48.8 ± 7.6	0.732	0.880	−4.29	6.05
	ACT	14	48.0 ± 7.8				
SF-36 Mental Sumscale	INACT	26	49.7 ± 10.5	0,413	2.764	−3.99	9.52
	ACT	14	46.9 ± 9.1				
SF-36 Change in Health Status	INACT	28	2.9 ± 0.7	0.547	0.143	−0.33	0.62
	ACT	16	2.8 ± 0.8				

INACT, Inactive Group; ACT, Active Group; ODI, Oswestry Disability Index; 30secSTS, 30 s Sit-to-Stand Test; GAD-7, Generalized Anxiety Disorder-7; WAI, Work Ability Index; PHQ-9, Patient Health Questionnaire-9; PSS-10, Perceived Stress Scale-10; SF-36, Short Form-36; BL, Baseline; FU, Follow-Up.

**Table 2 ijerph-21-00772-t002:** Mixed-models ANOVA: Effects of time and interaction effects between time and group allocation in all outcome parameters.

Outcome Parameter	Group	N	BL Mean ± SD	FU Mean ± SD	Effect of Time (df) F	Effect of Time *p*	Interaction Effect (df) F	Interaction Effect *p*
ODI	INACT	28	12.0 ± 8.3	8.9 ± 7.7	(1, 42) 7.042	0.011 *	(1, 42) 0.004	0.951
	ACT	16	13.5 ± 7.5	10.5 ± 7.3				
30secSTS	INACT	28	18.6 ± 4.1	20.4 ± 5.1	(1, 42) 12.409	0.001 *	(1, 42) 0.430	0.516
	ACT	16	18.6 ± 4.4	19.9 ± 5.9				
GAD-7	INACT	28	5.5 ± 5.9	3.7 ± 3.8	(1, 42) 4.656	0.037 *	(1, 42) 0.668	0.418
	ACT	16	7.1 ± 4.2	6.3 ± 4.3				
WAI	INACT	28	39.6 ± 7.6	41.4 ± 6.7	(1, 42) 2.811	0.101	(1, 42) 2.129	0.152
	ACT	16	35.3 ± 5.8	35.4 ± 6.0				
PHQ-9	INACT	28	5.0 ± 5.0	3.6 ± 3.1	(1, 42) 1.194	0.281	(1, 42) 1.194	0.281
	ACT	16	7.1 ± 4.9	7.1 ± 5.8				
PSS-10	INACT	28	12.3 ± 7.0	11.2 ± 7.3	(1, 42) 0.009	0.926	(1, 42) 1.833	0.183
	ACT	16	14.5 ± 6.9	15.8 ± 7.8				
SF-36 Physical Sumscale	INACT	26	48.3 ± 7.6	50.9 ± 6.5	(1, 38) 1.286	0.264	(1, 38) 0.255	0.617
	ACT	14	48.0 ± 7.8	48.7 ± 6.7				
SF-36 Mental Sumscale	INACT	26	49.7 ± 10.5	49.4 ± 10.2	(1, 38) 0.000	1.000	(1, 38) 0.051	0.822
	ACT	14	49.4 ± 10.2	47.2 ± 11.8				
SF-36 Change in Health Status	INACT	28	2.9 ± 0.7	2.6 ± 0.8	(1, 42) 0.592	0.446	(1, 42) 3.059	0.088
	ACT	16	2.8 ± 0.8	2.9 ± 0.5				
SF-36 Physical Functioning	INACT	27	85.9 ± 17.7	91.1 ± 9.6	(1, 41) 5.936	0.019 *	(1, 41) 0.001	0.977
	ACT	16	82.2 ± 11.8	87.5 ± 11.1				
SF-36 Role Physical	INACT	28	83.0 ± 30.5	87.5 ± 25.0	(1, 40) 0.057	0.812	(1, 40) 0.311	0.580
	ACT	14	78.6 ± 29.2	76.8 ± 31.7				
SF-36 Bodily Pain	INACT	28	69.2 ± 23.0	73.6 ± 21.8	(1, 42) 1.990	0.166	(1, 42) 0.009	0.925
	ACT	16	66.9 ± 20.4	72.0 ± 16.1				
SF-36 General Health	INACT	27	71.2 ± 16.7	71.0 ± 18.8	(1, 41) 1.649	0.206	(1, 41) 1.342	0.253
	ACT	16	62.8 ± 20.9	59.1 ± 21.0				
SF-36 Vitality	INACT	28	52.5 ± 19.1	56.7 ± 20.4	(1, 42) 1.423	0.240	(1, 42) 0.777	0.383
	ACT	16	48.8 ± 17.6	49.4 ± 21.5				
SF-36 Social Functioning	INACT	28	85.7 ± 20.6	88.4 ± 19.5	(1, 42) 0.095	0.759	(1, 42) 0.316	0.577
	ACT	16	80.5 ± 20.4	79.7 ± 19.3				
SF-36 Role Emotional	INACT	28	92.9 ± 26.2	85.7 ± 27.9	(1, 41) 0.340	0.563	(1, 41) 1.233	0.273
	ACT	15	80.0 ± 30.3	82.2 ± 33.0				
SF-36 Mental Health	INACT	28	72.6 ± 19.4	75.7 ± 17.9	(1, 42) 1.401	0.243	(1, 42) 0.529	0.471
	ACT	16	68.5 ± 17.5	69.3 ± 20.3				

INACT, Inactive Group; ACT, Active Group; ODI, Oswestry Disability Index; 30secSTS, 30 s Sit-to-Stand Test; GAD-7, Generalized Anxiety Disorder-7; WAI, Work Ability Index; PHQ-9, Patient Health Questionnaire-9; PSS-10, Perceived Stress Scale-10; SF-36, Short Form-36; BL, Baseline; FU, Follow-Up. * sig. effect.

## Data Availability

The data presented in this study are available on request from the corresponding author.

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
