# Peer review of "“Back Health 24/7/365”—A Novel, Comprehensive “One Size Fits All” Workplace Health Promotion Intervention for Occupational Back Health among Hospital Employees"

_ijerph, 2024, doi:10.3390/ijerph21060772_

Round 1
Reviewer 1 Report
Comments and Suggestions for Authors
The discussion could be wider. For example:
1) the reasons why only 0.76% of the GH total staff participated?
2) participation/dropout by occupation/job profile?
3) Impact on the INACT group was higher as they started from a lower level?
Reviewer 2 Report
Comments and Suggestions for Authors
1 - General Comment: I appreciate having had the opportunity to review this manuscript. The study under review presents a relevant theme in promoting worker health. This study was included in this perspective with the objective of the WPH project, which was to provide theoretical and practical training to hospital workers on maintaining a healthy spine during work and in all daily activities.
Introduction
2 - Comment: The Introduction is coherent. It presents the issue, justifying and arguing its regional, national, and global relevance using appropriate references. The objective is clear, and the nuclear issue is articulated in its hypotheses.
3 -Suggestion: In the Introduction, specific information about the study's location and population should be in the Materials and Methods section. In other words, the relevance of carrying out the present research in this location can be maintained; only the way of presenting it, including details about this service, can, I think, be left in the Materials and Methods section.
Materials and methods
4 – Comment: ​​Materials and Methods. I evaluate the research design as adequate and coherent with the object of investigation, sampling procedures, intervention design, and its evaluation and statistical analysis procedures.
5 - Comment: I also highlight the presentation of Supplementary Materials, which complement the results through complementary figures and a detailed description of the content of the WPH Project (intervention process) carried out, which played a crucial role in the study.
Results:
6 – Suggestion: I recommend that particular medical reasons are not provided, that is, only for medical reasons. It also indicates, for instance, which questionnaire one of the guests considered "too personal." Furthermore, I suggest that the authors try to write in a way that removes quotation marks ("); I understand that this becomes more formal.
7 - Suggestion: I recommend that additional figures be indicated in the text throughout the results section, for example: (Additional File Figure S1...) or (Supplement Material Figure S1...). These are just examples; authors can make indications according to IJERPH presentation rules. Furthermore, authors may also choose to present some of the Figures in the results text.
Discussion
8 – Comment: The discussion is summarized adequately.
Conclusions
9 – Suggestion: The conclusions are also relevant to indicate secondary results related to the primary outcome.
References:
10 - Comment: The indicated references are appropriate to the issue and contribute to the scientific argumentation of the content developed in the manuscript.
I am grateful for the opportunity to review this manuscript and wish the team success.
